# FLAG: CLUSTERED FEDERATED LEARNING COMBINING DATA AND GRADIENT INFORMATION IN HETEROGENEOUS SETTINGS

## ABSTRACT

Federated Learning (FL) emerged as an important tool to enable a group of agents/clients to collaboratively train a model without sharing their individual data with each other or any third party, instead exchanging only model updates during each training round. Although FL performs effectively when clients' data are homogeneous (e.g., each client's data is i.i.d.), data heterogeneity among clients presents a major challenge, often leading to significant performance degradation. To address this challenge, a variety of approaches have been proposed. One particularly effective approach is clustered FL, where similar clients are grouped together to train separate models. Previous clustered FL approaches tend to rely solely on either data similarity or gradient similarity to cluster clients. This results in an incomplete assessment of client similarities, particularly when the datasets display various types of distributional skews, such as label, feature, or quantity imbalances. Consequently, these methods fail to capture the full spectrum of client heterogeneity, leading to suboptimal model performance across diverse client environments.

In this work, we address the challenge of data heterogeneity in FL by introducing a novel clustered FL approach, called FLAG. FLAG employs a weighted class-wise similarity metric that integrates both data and gradient similarity, providing a more holistic measure of client similarity. This enables more accurate clustering of clients, ultimately improving model performance across heterogeneous data distributions. Our extensive empirical evaluation on multiple benchmark datasets, under various heterogeneous data scenarios, demonstrates that FLAG consistently outperforms state-of-the-art approaches in terms of accuracy.

## 1 INTRODUCTION

Federated Learning (FL) enables users/clients to collaboratively train a model on their data without sharing it with other clients or a central entity (McMahan et al., 2017). However, the diversity in user behavior often results in heterogeneous data distributions, referred to as *non-identically independently distributed* (non-IID) data, across clients. This heterogeneity can lead to slower convergence and suboptimal accuracy of the global model (Kairouz et al., 2021; Tan et al., 2022). For example, in disease risk prediction using electronic health records, variations in patient demographics and clinical presentations contribute to data heterogeneity (Prayitno et al., 2021). More specifically, non-IID data can arise due to various factors, including class/label skew, feature skew, quantity shift, concept shift, and concept drift — common types of data heterogeneity. *Class/label skew* refers to the non-identical distribution of labels/classes at different clients, e.g., the absence of a label at one client while the same label is present at other clients (Zhang et al., 2022). *Feature skew* occurs when distributions vary due to different personalization nuances, e.g., an alphabet letter can be written in different ways (Li et al., 2021). *Quantity shift* happens when different clients have different amounts of data (Wang et al., 2021), e.g., an online retailer with millions of transaction records is compared to a local store with only a few hundred records. *Concept drift* occurs when the statistical properties of the target variable change, leading to different labels for similar data instances across clients (Kairouz et al., 2021). *Concept shift* happens when different clients assign the same label to fundamentally different data samples due to variations in local data distributions or labeling criteria (Kang et al., 2024).

Approaches, e.g., personalized FL (Fallah et al., 2020; Liang et al., 2020; Smith et al., 2017; Arivazhagan et al., 2019), aggregation schemes (Wang et al., 2020; Pillutla et al., 2022; Karimireddy et al., 2020), local-global mixing (Jiang et al., 2024; Mansour et al., 2020; Deng et al., 2020), and clustered FL (Ghosh et al., 2020; Vahidian et al., 2023; Sattler et al., 2020; Long et al., 2023), have been proposed to address data heterogeneity. Personalized FL and aggregation-based approaches, which train a single model, often fail to generalize well to the local distributions of each client. In local-global mixing approaches, where local distributions differ significantly from the global average, these methods deteriorate, causing each client to train primarily on its own local data (Vahidian et al., 2023).

Clustered FL, in contrast, has demonstrated superior performance in handling non-IID data, especially when distinct groups of clients display substantial variations in their local data distributions (Ghosh et al., 2020). In clustered FL, clients are grouped into clusters based on the similarities in their data distributions, and each cluster trains its own model tailored to its specific data.

Despite their advantages, existing clustered FL approaches face several limitations, such as limited flexibility in capturing complex data heterogeneity, reliance on either data or gradient similarity alone. Specifically, current clustered FL approaches suffer from the following limitations:

1. **Data-only or Gradient-only Clustering:** Existing clustered FL approaches rely on either data subspace (Vahidian et al., 2023) or gradient subspace to compute similarity (Sattler et al., 2020) between clients.In gradient-only similarity measurement, nearly similar clients can end up with different learning objectives due to class imbalance (i.e., quantity shift) or high dimensionality. In data-only cases, clients may exhibit intra-class variance, concept shift, or concept drift, which may hinder similarity measurement.

2. **Improper Data Similarity Method:** Data similarity-based clustered FL methods, such as PACFL (Vahidian et al., 2023), uses a method that leverages cosine similarity measurement between data subspace but does not account for label information of the data subspace during comparison. In cases of concept shift, data subspace can exhibit similarity while having different labels, which might result in incorrect similarity assessment.

3. **Predefined Cluster Numbers**: Clustered FL approaches, such as IFCA (Ghosh et al., 2020), assume that the number of clusters are known before running FL training, which can lead to poor model performance, if the predefined number does not match the actual data distributions. Previous approaches lack a mechanism for determining the optimal number of clusters when it is not known in advance.

4. **Limited Consideration of Data Skews:** Existing clustered FL techniques primarily focus on experiments with one or two types of data distribution skews, predominantly class/label skew. They do not account for the broader range of skews, such as concept shift, concept drift, and quantity shift, in data heterogeneity.

The above-mentioned limitations raise the following crucial question:

*How can we overcome the above challenges posed by heterogeneous data distributions by utilizing both data and gradient information to dynamically group clients into clusters in FL ?*

**Our Contributions.** This work proposes a novel algorithm, titled clustered Federated Learning with datA and Gradient (FLAG), which integrates both data and gradient information to group clients. FLAG effectively addresses the limitations of existing clustered FL methods. By combining gradient and data information with an enhanced similarity measurement in data-space, FLAG tackles a wide range of data heterogeneity challenges, resolving the key limitations of current approaches. Specifically, in FLAG, each client first performs truncated Singular Value Decomposition (SVD) (Klema & Laub, 1980) on their dataset and sends a few principal vectors to the server. Furthermore, the client executes a few epochs of local training on their data using Stochastic Gradient Descent (SGD) (Ruder, 2016) and sends the local model gradients to the server.

The server uses the principal vectors to build a weighted class-wise data similarity matrix for the clients. In addition, the server computes the cosine angle between the gradient directions of the clients and constructs a gradient similarity matrix based on those values. Finally, the server combines these two, i.e., data and gradient, matrices to create the final proximity matrix, which is used as the adjacency/distance matrix for clustering clients using Agglomerative Hierarchical Clustering (HC).

FLAG also employs an approach to find the optimal clustering. It does so by first iterating over the clustering threshold in hierarchical clustering (HC), a hyper-parameter that controls the number of

Table 1: Test accuracy comparison across different clustering algorithms for non-IID label skew (20%) and quantity shift (Dirichlet concentration factor 1) over CIFAR-10 and FMNIST data.

| ALGORITHM | TECHNIQUE | CIFAR-10 | FMNIST |
|---|---|---|---|
| PACFL Vahidian et al. (2023) | Data | 90.45 | 94.412 |
| CFL Sattler et al. (2020) | Gradient | 72.80 | 86.973 |
| IFCA Ghosh et al. (2020) | Gradient | 89.68 | 94.027 |
| **FLAG — THIS PAPER** | **DATA+GRADIENT** | **93.81** | **96.36** |

clusters. It performs clustering only once before training begins, eliminating the need to wait until iterative training completes to form clusters, as is required in many previous clustered FL frameworks. FLAG selects the best clustering based on the validation accuracy and performs clustered FL on that optimal clustering.

In summary, the contributions of this paper are as follows:

1. A new algorithm, FLAG, for clustered federated learning that combines both data and gradient information from clients to cluster them into groups.
2. A novel class-wise weighted method to compute similarity in data and capture the variations in underlying distributions of the clients' data more accurately and robustly in different heterogeneous scenarios with skews.
3. An efficient and empirical way to determine the optimal clustering formation for the clients.
4. Extensive experimentation on heterogeneous data distributions was conducted, considering not only class types but also quantity shifts and class imbalances among clients working with the same class types. Table 1 provides an overview of the accuracy of the FLAG algorithm compared to existing clustered FL algorithms for non-IID data with class skew and label skew combined (details of FL setup in §5.1), showing its effectiveness.

**Code and data** of the paper is provided here URL.

## 2 LITERATURE REVIEW

Clustered federated learning (FL) is a technique that addresses the distribution shift problem by grouping clients into clusters based on their local data distributions. Various methods have been proposed for clustering clients. Vahidian et al. (2023) introduced a method that clusters clients by analyzing the principal angles between the client data subspaces. However, this approach does not account for the label information of the datasets being compared. In scenarios such as concept shift and concept drift, datasets with similar subspaces can have different labels and learning objectives. Ignoring label information may lead to incorrect clustering of clients. Additionally, the method does not regularize the similarity values based on dataset size to address quantity shifts. Another type of clustered FL approach leverages gradient information or loss values on gradient to cluster clients (Kim et al., 2024). Ghosh et al. (2020) proposed initializing a set of global models and, at each round, estimating the cluster identity of the clients based on the minimum loss to the global model parameters. Then, clients with similar identities are aggregated to minimize the loss function. However, this method assumes the number of clusters is known beforehand and performs clustering (estimating identities) of clients at each iteration, making it computationally expensive. Another gradient-based clustering approach recursively performs bi-partitioning when clients' gradients differ on a converged global model (Sattler et al., 2020). Zhang et al. (2024) developed an adaptive clustering algorithm based on cosine-based model similarity of dimensionality-reduced models. Ruan & Joe-Wong (2022) proposed an approach that employs soft clustering instead of hard clustering, utilizing a proximal local updating technique that incorporates local information while encoding knowledge from all cluster models. Another soft clustering-based approach, Guo et al. (2023), formulates the clustering problem as a bi-level optimization problem and introduces a new objective function to achieve robust clustering.

## 3 PRELIMINARIES

**Principal Angles Between Two Subspaces.** Let $\text{span}\{\mathbf{v}_1, \ldots, \mathbf{v}_p\}$ be the span of a set of vectors and denotes the set of all possible linear combinations of these vectors. Let $\mathbb{R}^n$ be the $n$-dimensional real coordinate space, which is the set of all $n$-tuples of real numbers. Let $\mathcal{V} = \text{span}\{\mathbf{v}_1, \ldots, \mathbf{v}_p\}$ and $\mathcal{X} = \text{span}\{\mathbf{x}_1, \ldots, \mathbf{x}_q\}$, be $p$-dimensional and $q$-dimensional subspaces of $\mathbb{R}^n$, respectively.

The sets $\{\mathbf{v}_1, \ldots, \mathbf{v}_p\}$ and $\{\mathbf{x}_1, \ldots, \mathbf{x}_q\}$ are orthonormal, with $1 \leq p \leq q$. We define a sequence of $p$ principal angles as $0 \leq \Phi_1 \leq \Phi_2 \leq \cdots \leq \Phi_p \leq \frac{\pi}{2}$, and they measure the similarity between the subspaces. In this context, principal angles are the angles between the closest directions in two subspaces, and they provide a measure of how "aligned" or "separated" two subspaces are in space. These angles are calculated as:

$$\Phi(\mathcal{V}, \mathcal{X}) = \min_{\mathbf{v} \in \mathcal{V}, \mathbf{x} \in \mathcal{X}} \cos^{-1}\left(\frac{|\mathbf{v}^T \mathbf{x}|}{\|\mathbf{v}\|\|\mathbf{x}\|}\right) \quad (1)$$

where $\|\cdot\|$ is the norm and $\mathbf{v}^T$ represents the transpose of the matrix $\mathbf{v}$. The smallest of these angles is $\Phi_1(\mathbf{v}_1, \mathbf{x}_1)$, with the vectors $\mathbf{v}_1$ and $\mathbf{x}_1$ as the corresponding principal vectors. The principal angle distance serves as a metric to quantify the separation between subspaces (Jain et al., 2013).

**Gradient based clustering** cluster clients based on their gradient similarity (e.g., Duan et al. (2021); Sattler et al. (2020)) or use loss value on gradient update to identify similar clients (Ghosh et al., 2020). Each client $i \in N$ is initialized with a random model parameter $\theta$ and trains the model on its local data $D^i$ until it converges to a stopping point. Afterward, each client $i$ sends their *local* gradient updates (denoted by $\Delta^i$) to the server for similarity computation. The norm of the client's gradient $\|\Delta^i\|$ will tend toward zero as we are approaching the stationary point (Drori & Shamir, 2020). So, we can define the stopping point of clients by setting a threshold on the norm of the gradient update, such that $\|\Delta^i\| < \epsilon$. Another way to set up a stopping point that is used in FLAG is to train each local model for a predefined number of epochs. Upon receiving all the gradient updates, the server derives the gradient similarity value $\mathcal{G}_{i,j}$ between any two clients $i$ and $j$ by computing the cosine angle between the two gradient updates $\Delta^i$ and $\Delta^j$; see Eq. 2. A cosine angle value $\theta_{i,j} = 0°$ implies perfect alignment, $\theta_{i,j} = 90°$ indicates orthogonality, and $\theta_{i,j} = 180°$ means the vectors are diametrically opposed.

$$\mathcal{G}_{i,j} = \theta_{i,j} = \cos^{-1}\left(\frac{\langle \Delta^i \cdot \Delta^j \rangle}{\|\Delta^i\|\|\Delta^j\|}\right) \times \frac{180}{\pi} \quad (2)$$

**Agglomerative hierarchical clustering** (Day & Edelsbrunner, 1984) is a popular method in machine learning for grouping similar objects based on a proximity (or adjacency) matrix. The process begins by treating each data point as its own cluster. During each iteration, the algorithm carries out two main tasks: identifying the two clusters that are most similar and merging these clusters. The criterion for selecting which clusters to merge depends on a linkage method, such as single, complete, or average linkage. For instance, in *single linkage*, the $L_2$ (Euclidean) distance between two clusters is defined as the smallest distance between any pair of points from the two clusters. For merging criteria, FLAG defines ***clustering threshold*** as $\alpha$, i.e., any two clusters with a distance less than $\alpha$ are merged together; see details in §4.2.

## 4 FLAG ALGORITHM

This section develops FLAG algorithm (see Algorithm 1) that is based on weighted class-wise data similarity among clients (Algorithm 2) and combining both data and gradient to obtain better clustering (Algorithm 3). Server computes the optimal clustering during the first iteration and performs the clustered FL in the subsequent iterations. The components of our approach are described below.

### 4.1 WEIGHTED CLASS-WISE DATA BASED SIMILARITY

**Objective and high-level idea.** Our first objective is to find the similarity among the client based on their data. For this, FLAG uses cosine similarity between data of the same class for each pair of clients and then takes the average across all the classes, which produces the proper magnitude of similarity instead of just a binary similarity outcome. To account for the quantity shift between the classes of the two participating clients, we assign weights to the class similarity values based on the difference in class frequency, as discussed below in detail.

**Details of the method.** Let $C$ be the number of classes and $N$ be the number of clients. A client can have $c \leq C$ classes. Let $D_{i,c}$ be data at client $i$ for class $c$ in the form of a matrix, where rows refer to data points, and columns refer to features. The process starts at the client by computing a set of *principal vectors* for each class. The principal vectors are a linear combination of the actual data and are sent to the server, which computes the principal angle between each pair of clients, which

will serve as the basis for computing the weighted class-wise data similarity matrix. The number of principal vectors sent to the server is less than 1% in size of the actual class data. The client applies truncated SVD (Klema & Laub, 1980)[1] on the transpose of $D_{i,c}$.[2] This results in $p$ principal vectors for each $c$ class at the client $i$, denoted by $U_c^i = [u_1, u_2, \ldots, u_p]$. We take small values for $p$ to keep the number of principal vectors minimal. After that, each client sends their $U_c^i$ to the server. For each pair of clients $i$ and $j$, the server computes the principal angle between $U_c^i$ and $U_c^j$, as discussed in §3, for the same class $c$, resulting in a *principal angle matrix*, $\mathcal{V}'_{i,j,c}$, see Eq. 3:

$$\mathcal{V}'_{i,j,c} = \begin{cases} \Phi(U_c^i, U_c^j), & \text{if } c \text{ is in both } U^i \text{ and } U^j, \\ 180°, & \text{if } c \text{ in either } U^i \text{ or } U^j, \qquad i,j = 1,\ldots,N; \ c = 1,\ldots,C \\ 0°, & \text{if } c \text{ in neither } U^i \text{ nor } U^j, \end{cases} \tag{3}$$

The smaller the value of the principal angle, the more similar the data of class $c$ is between clients $i$ and $j$. If the class $c$ is absent in one of $U^i$ or $U^j$, we assign $180°$ to account for the dissimilar subspace, and $0°$ when $c$ is absent in both datasets to exhibit similarity.

Next, the server derives the similarity score to build the ***weighted class-wise data similarity matrix***, denoted by $\mathcal{V}_{i,j}$. To do so, the server takes the weighted average of all principal angles across all the classes in each pair of clients $i$ and $j$. The weighting scheme ensures that significant differences in data size between clients $i$ and $j$ for class $c$ will increase the class principal angle to reflect greater dissimilarity, while smaller differences will increase similarity. The server first computes the weight for the principal angle of each class $c$ between client $i$ and $j$ using Eq 4, resulting in a *weight matrix* $\mathcal{W}_{i,j,c}$, where $|D_{i,c}|$ is the size of the data of client $i$ for class $c$. and $\epsilon$ is a small positive value introduced to ensure numerical stability.

$$\mathcal{W}_{i,j,c} = \frac{\max(\ln(|D_{i,c}| + \epsilon), \ln(|D_{j,c}| + \epsilon))}{\min(\ln(|D_{i,c}| + \epsilon), \ln(|D_{j,c}| + \epsilon))} \quad i,j = 1,\ldots,N; \ c = 1,\ldots,C \tag{4}$$

Then, the server normalizes the weights (see Eq 5), ensuring the weight lies in a given range of $[1-\delta, 1+\delta]$, where $\delta$ is a positive constant that can be regularized by the server based on the importance of weights. Here, $w_{min}$ (or $w_{max}$) refers to the minimum (or maximum) weight value in the weight matrix $\mathcal{W}_{i,j,c}$.

$$\mathcal{W}'_{i,j,c} = (1 - \delta) + \frac{(\mathcal{W}_{i,j,c} - w_{min})((1 + \delta) - (1 - \delta))}{w_{max} - w_{min}} \tag{5}$$

Finally, the server multiplies the principal angles by its corresponding normalized weights to compute *weighted class-wise data similarity matrix* $\mathcal{V}_{i,j}$, see Eq. 6.

$$\mathcal{V}_{i,j} = \frac{1}{|C|} \sum_{c=1}^{C} \mathcal{V}'_{i,j,c} \times \mathcal{W}'_{i,j,c} \tag{6}$$

## 4.2 Optimal Clustering & Combining Data and Gradient — Algorithms 2 and 3

**Objective and high-level idea.** Once the server computes weighted class-wise similarity matrix $\mathcal{V}_{i,j}$ (as discussed in §4.1), our objective is to identify the optimal cluster formation. For this, FLAG combines data and gradient information. First, the server computes a *proximity matrix* based on $\mathcal{V}_{i,j}$ and a gradient similarity matrix $\mathcal{G}_{i,j}$ and then uses the proximity matrix as the adjacency matrix for agglomerative hierarchical clustering (HC) (Murtagh & Contreras, 2012) to cluster the clients together.

**Details of the method.** To compute the gradient, each client $i$ performs a few rounds ($t_g$) of training using stochastic gradient descent (SGD) only on its local data, and computes its local gradient direction $\Delta^i$ and sends this gradient direction to the server. The server computes the gradient similarity matrix $\mathcal{G}_{i,j}$ by calculating the cosine similarity between the gradient directions, as discussed in §3.

**Proximity Matrix $\mathcal{A}_{i,j}$ (Algorithm 2).** The server requires the proximity/adjacency matrix for the clustering algorithm. First, the server normalizes $\mathcal{V}_{i,j}$ and $\mathcal{G}_{i,j}$ matrices using min-max normalization (Patro, 2015) to maintain consistency of scale, resulting in $\hat{\mathcal{V}}_{i,j}$ and $\hat{\mathcal{G}}_{i,j}$. Then, the server combines them to obtain the *proximity/adjacency matrix*. The reason for combining $\hat{\mathcal{V}}_{i,j}$ and $\hat{\mathcal{G}}_{i,j}$

---

[1]Truncated SVD is a variation of SVD where only a subset of the singular values and corresponding singular vectors are computed. SVD shows a good trade-off between computational efficiency and the representational quality of subspace methods (Talwalkar et al., 2013).

[2]We apply SVD on transposed $D_{i,c}$ because we want to compute principal vectors, not principal features.

is as follows: using only one factor (data or gradient) to compute client similarity can result in inaccurate clustering; particularly, in datasets with skews, for example, gradient-only measures can misalign clients' learning objectives due to class imbalance or high dimensionality, while data-only approaches may struggle with intra-class variance, concept shift, or drift. Combining both data and gradient information helps mitigate these issues, leading to more accurate similarity measurements. The server computes the proximity matrix as: $\mathcal{A}_{i,j} = \beta \cdot \hat{\mathcal{V}}_{i,j} + (1-\beta) \cdot \hat{\mathcal{G}}_{i,j}$. Here, $\beta$ is the parameter to control the weight ratio of each element.

**Finding Optimal Clustering (Algorithm 3).** Given the proximity/adjacency matrix $\mathcal{A}_{i,j}$, the server uses a hierarchical clustering (HC) (Murtagh & Contreras, 2012) to find the best clustering. HC works by finding the most similar clusters and merging them together. In HC, *clustering threshold*, $\alpha \in (0, 1]$ serves as a merging criterion for any pair of clusters, meaning clusters with a distance smaller than the threshold are eligible for merging; $\alpha$ also controls the number of clusters formed. For example, $\alpha=1$ results in all clients being grouped into a single cluster. The server uses $\mathcal{A}_{i,j}$ as a distance metric for HC and iterates over different values of the $\alpha$ to find the best clustering. The server systematically decreases $\alpha$ from 1 to 0 in regular intervals (e.g., 0.1) to generate different clustering configurations. To assess the goodness or performance of each generated clustering without executing the full clustered FL training, FLAG uses an efficient alternative method, as follows: the server selects only a few clients, and those clients set aside $10\%$ of their local data as a validation set. Then, with the remaining data, the selected clients train an efficient lightweight model with simple architecture in the Clustered FL setting based on that corresponding clustering. After training the model for a few epochs, each client evaluates the model's accuracy on the validation set and sends the accuracy score to the server. The server computes the average accuracy across all the selected clients, which serves as the cluster goodness metric for the corresponding clustering.

Finally, the server determines the clustering goodness metrics for each $\alpha$ values and uses methods such as the elbow method (Syakur et al., 2018) or some other criteria to select the best $\alpha$ and the corresponding optimal clustering $\{\mathbb{C}_1, \ldots, \mathbb{C}_Z\}$, where $Z$ is the number of clusters.

## 4.3 CLUSTERED FEDERATED LEARNING

**Objective and high-level idea.** This section describes how the server utilizes the previously received clustering and run clustered FL to obtain the updated global models. The server performs clustered federated learning by maintaining separate models for each cluster and aggregating the locally trained models received from clients within the clusters.

**Details of the method.** At this stage, after obtaining the optimal clustering $\{\mathbb{C}_1, \ldots, \mathbb{C}_Z\}$ (as discussed in §4.2), the server performs the clustered FL. Since FLAG performs one-shot clustering in the first iteration, the server does not need to perform clustering in subsequent iterations. For clustered FL, the server maintains a separate model for each cluster $z \in Z$ and initializes each with a random model parameter $\theta_{g,z}^0$; where $\theta_{g,z}^0$ denotes the cluster $z$ initial global model parameter for iteration $t=0$. In each round $t$, the server samples a set of available clients $S_t = \{i_1, i_2, \ldots, i_m\}$ from the client population, where $m$ is determined by the sampling rate $R \in (0, 1]$. The server then broadcasts the cluster-specific global model parameters $\theta_{g,z}^t$ to each client $i \in \mathbb{C}_z$ corresponding to its cluster assignment. Upon receiving the model, each client $i$ performs local training on its own dataset $D_i$, optimizing the received model $\theta_{g,z}^t$ through stochastic gradient descent (SGD) for a fixed number of local epochs. After local training, each client $i$ computes an updated local model $\theta_{i,z}^{t+1}$ and sends it back to the server. This process is expressed as:

$$\theta_{i,z}^{t+1} \leftarrow \text{ClientUpdate}(i, \theta_{g,z}^t) \tag{7}$$

where ClientUpdate denotes the model update with SGD using the client's data. Once all selected clients have sent their locally updated models, the server aggregates the clients' models by performing weighted model averaging within each cluster The weighted average for each cluster $z$ is computed as follows:

$$\theta_{g,z}^{t+1} \leftarrow \sum_{i \in C_z} \frac{|D_i| \theta_{i,z}^{t+1}}{\sum_{i \in C_z} |D_i|} \tag{8}$$

where $|D_i|$ represents the size of client $i$'s local dataset. The server then repeats the process to continue the clustered FL training process (see Lines 14-16 of Algorithm 1 ).

---

**Algorithm 1:** FLAG ALGORITHM

**Input:** Number of clients $N$, sampling rate $R \in (0, 1]$, combination ratio $\beta$, clustering threshold $\alpha$, number of classes $C$
**Output:** Updated global model parameters

1  Initialize the server model with $\theta_g^0$;
2  **for** *each round* $t = 0, 1, \ldots$ **do**
3  $\quad$ $m \leftarrow \max(R \cdot N, 1)$ // Sampling rate
4  $\quad$ $S_m \leftarrow \{i_1, \ldots, i_m\}$ // Set of $m$ sampled clients
5  $\quad$ **for** *each client* $i \in N$ **in parallel do**
6  $\quad\quad$ **if** $t = 0$ *(For one-shot clustering)* **then**
7  $\quad\quad\quad$ Client $i$ applies Singular Value Decomposition (SVD) and extracts $U^i$ for each class $c \in \mathbb{C}$ and sends to server
8  $\quad\quad\quad$ Client $i$ trains on local data for $t_g$ epochs and sends $\Delta^i$ to the server
9  $\quad\quad\quad$ Server forms the proximity matrix $\mathcal{A} \leftarrow \texttt{ProximityMatrix}(U^*, \Delta^*, \beta)$
$\quad\quad\quad\quad$ (Algorithm 2)// $U^* = \{U^1, \ldots, U^N\}$, $\beta$ is combination ratio of data and gradient
10 $\quad\quad\quad$ Clusters $\mathbb{C}_1, \ldots, \mathbb{C}_Z \leftarrow \texttt{OptimalClusteringSearch}(\mathcal{A})$ (Algorithm 3)// Find optimal clustering
$\quad\quad\quad\quad$ using the threshold search function
11 $\quad\quad\quad$ Initialize all cluster models $\theta_{g,z}^0 \leftarrow \theta_g^0$ for all $z \in Z$
12 $\quad\quad$ **else**
13 $\quad\quad\quad$ Client $i$ (if sampled in $S_m$) receives the corresponding cluster model $\theta_{g,z}^t$ from the server;
14 $\quad\quad$ $\theta_{i,z}^{t+1} \leftarrow \texttt{ClientUpdate}(i, \theta_{g,z}^t)$ // if client $i$ sampled in $S_m$
15 $\quad$ **for** *each cluster* $z = 1$ *to* $Z$ **do**
16 $\quad\quad$ $\theta_{g,z}^{t+1} \leftarrow \sum_{i \in C_z} \frac{|D_i|\theta_{i,z}^{t+1}}{\sum_{i \in C_z} |D_i|}$ // Apply model averaging for each cluster
17 **Function** $\texttt{ClientUpdate}(i, \theta_{g,z}^t)$:
18 $\quad$ With Stochastic Gradient Descent (SGD) train local model on client $i$'s data to get $\theta_{i,z}^{t+1}$
19 $\quad$ **return** $\theta_{i,z}^{t+1}$

---

**Algorithm 2:** Proximity Matrix Calculation for Clustering

**Input:** The set of first $p$ principal vectors for all clients $U^*$, all clients' local model gradient direction vectors $\Delta^*$
**Output:** $\mathcal{A}$, proximity matrix between all pairs of clients for clustering.

1  **Function:** $\texttt{ProximityMatrix}(U^*, \Delta^*, \beta)$
2  **for** *client* $i = 1, 2, \ldots, N$ **do**
3  $\quad$ **for** *client* $j = i, 2, \ldots, N$ **do**
4  $\quad\quad$ **for** *class* $c = 1, 2, \ldots, C$ **do**
5  $\quad\quad\quad$ $\mathcal{V}'_{i,j,c} \leftarrow$ execute Eq. 3 and $\mathcal{W}_{i,j,c} \leftarrow$ execute Eq. 4
6  $\mathcal{W}' \leftarrow$ normalize weight matrix $\mathcal{W}$ as per Eq. 5
7  **for** *client* $i = 1, 2, \ldots, N$ **do**
8  $\quad$ **for** *client* $j = 1, 2, \ldots, N$ **do**
9  $\quad\quad$ $\mathcal{V}_{i,j} \leftarrow \frac{1}{|C|} \sum_{c \in C} \mathcal{V}'_{i,j,c} \times \mathcal{W}'_{i,j,c}$ // Data-based similarity matrix, averaged over all classes
10 $\quad\quad$ $\mathcal{G}_{i,j} \leftarrow \cos^{-1}(\Delta^i \cdot \Delta^j)$ // Gradient based similarity matrix
11 $\hat{\mathcal{V}} \leftarrow$ normalize$(\mathcal{V})$, $\quad \hat{\mathcal{G}} \leftarrow$ normalize$(\mathcal{G})$ // Min-max normalize
12 **return** $\mathcal{A}_{i,j} \leftarrow$ normalize$(\mathcal{A}_{i,j}) \leftarrow \hat{\mathcal{V}} \cdot \beta + \hat{\mathcal{G}} \cdot (1 - \beta)$ // Final normalized proximity matrix combining $\mathcal{V}$ and $\mathcal{G}$

---

**Algorithm 3:** Function for Clustering Threshold Search in Federated Learning

**Input:** Proximity matrix $\mathcal{A}$
**Output:** Optimal clustering $\{\mathbb{C}_1, \ldots, \mathbb{C}_Z\}$

1  **Function** $\texttt{OptimalClusteringSearch}(\mathcal{A})$:
2  $\quad$ $S_{m'} \leftarrow \{i_1, \ldots, i_{m'}\}$ Select $m'$ clients at random from $N$
3  $\quad$ **for** *each client* $i \in S_{m'}$ **do**
4  $\quad\quad$ Client $i$ sets aside 10% of local data as validation set $V_i$ and keeps 90% as training data $T_i$
5  $\quad$ Initialize empty list to store clustering goodness metric $\mathcal{X} \leftarrow []$ // store average validation acc
6  $\quad$ **for** $\alpha \leftarrow \{1, 0.9, \ldots, 0.1\}$ **do**
7  $\quad\quad$ Generate candidate clustering $\mathbb{C}^\alpha$ using hierarchical clustering (HC) on $\mathcal{A}$ with threshold $\alpha$
8  $\quad\quad$ **for** *each client* $i \in S_{m'}$ **do**
9  $\quad\quad\quad$ Client $i$ trains a lightweight model on $T_i$ for $t'$ rounds in the clustered (FL) setting on $\mathbb{C}^\alpha$
10 $\quad\quad\quad$ Client $i$ evaluates the model on its validation set $V_i$
11 $\quad\quad\quad$ Client $i$ sends its validation accuracy score $A_i$ to the server
12 $\quad\quad$ Server computes the average validation accuracy $G_\alpha \leftarrow \frac{1}{|S_{m'}|} \sum_{i \in S_{m'}} A_i$ and append $G_\alpha$ to $\mathcal{X}$
13 $\quad$ Plot clustering goodness metrics $\mathcal{X}$ against corresponding $\alpha$ values
14 $\quad$ Use the elbow method or any other criteria to select the optimal threshold $\alpha^*$
15 $\quad$ **return** Optimal clustering $\{\mathbb{C}_1, \ldots, \mathbb{C}_Z\}$ corresponding to $\alpha^*$

---

# 5 EXPERIMENTS

This section provides details on the experiments and compares FLAG against existing work. We investigate the following questions:

1. how much better FLAG is compared to existing method — §5.1
2. how effectively does FLAG find clustering — §5.2

3. to what extent combining data and gradient improves upon using them separately — §5.3

4. how many communication round does FLAG take to converge—§5.4

**Datasets.** We used four popular datasets for the image classification task in the federated learning setting, i.e., CIFAR-10 (Krizhevsky et al., 2009), FMNIST (Xiao et al., 2017), SVHN (Netzer et al., 2011), CIFAR-100 (Krizhevsky et al., 2009).

**Baselines.** We compare FLAG against SOTA methods: (i) Single model methods: FedAvg (McMahan et al., 2017), FedProx (Li et al., 2020), (ii) personalized FL method: PerFedAvg (Fallah et al., 2020), (iii) clustered FL — data-based methods: PACFL(Vahidian et al., 2023), (iv) clustered FL — gradient-based method: IFCA (Ghosh et al., 2020), (CFL) (Sattler et al., 2020), FedSoft (Ruan & Joe-Wong, 2022).

**Setup.** Our experiments consider a total of 100 clients, with $20\%$ randomly selected for each round. Unless mentioned otherwise, the experiments run for 200 communication rounds, where each client performs 10 local epochs using a batch size of 10 and SGD as the local optimizer. For $U_c^i$, the principal vector sent per class is approximately 1% of $|D_{i,c}|$. For computing $\mathcal{G}$, each client runs 20 local epochs. For combination ratio between data gradient, $\beta$, we used grid-search to fine-tune through various values and found $\beta = 0.5$ generalizes more effectively across different heterogeneity. For finding the clustering threshold (Algorithm 3), the server selects 30 clients at random and runs 5 communication rounds of clustered FL.

**Creation of non-IID data.** We combine class skew with quantity shift to test our algorithm against other baselines. To simulate class skew, we first randomly select $\rho\%$ of the total available labels and assign that set of labels to a random set of clients. Then, we pick another set of clients and repeat the process until all clients are assigned $\rho\%$ of the labels. Next, we use the Dirichlet distribution (Ng et al., 2011) to distribute the samples of each label amongst the clients assigned to those labels. The Dirichlet distribution introduces quantity shift and class imbalance among clients, and also simulates varying levels of heterogeneity. Thus, we create a more realistic distribution, in which we have different groups of clients that work on different types of classes. And, clients working on similar types of classes exhibit quantity shift and class imbalance among them. An example of this would be predictive text input (discussed in §1), or e-commerce recommendation system, where different vendors sell different types of products (e.g., electronics, home goods), with some overlapping products, exhibiting class skew. However, each vendor may sell certain categories in higher volumes (e.g., one vendor focuses more on smartphones, another on laptops), thus creating class imbalance and quantity shift among clients handling same type of classes. Another such example include: healthcare diagnosis system with different clinics working with different disease cases.

## 5.1 LABEL SKEW AND QUANTITY SHIFT

In our experiments, we combined label skew and quantity shift for the non-IID data to test our algorithm. We have class skew $\rho = 20\%$ and $30\%$, and the Dirichlet distribution concentration parameter (denoted as $\alpha'$) is set to *1 for a low degree*, and to *0.25 for a high degree* of quantity shift non-IID.[3] Table 2 exhibits the performance comparison results among different algorithms for $\rho = 20\%$ and $30\%$ with $\alpha' = 1$ and Table 3 shows for $\rho = 20\%$ and $30\%$ with $\alpha' = 0.25$. We can observe that single global model-based FL baselines, i.e., FedAvg and FedProx, perform inefficiently due to weight divergence and model drift issues in heterogeneous settings (Zhao et al., 2018). From Table 2 and Table 3, we can observe that clustered FL methods (except CFL) yield better performance compared to other categories of FL approaches. It is also evident that FLAG consistently outperforms all SOTA algorithms across all datasets. This is because FLAG can effectively identify the number of underlying groups of clients that work on similar types of classes in the Non-IID data distribution. FLAG has outperformed all clustered FL approaches, including both data-based (e.g., PACFL) and gradient-based (e.g., CFL, IFCA) ones.

## 5.2 FINDING THE OPTIMAL CLUSTER FORMATION

The server uses hierarchical clustering (HC) (Murtagh & Contreras, 2012) to find the best clustering for the FL training. The server systematically decreases the clustering threshold $\alpha$ from 1 to 0 in regular intervals of 0.1 to generate different clustering configurations. To assess the quality of

---

[3]Higher values of the Dirichlet parameter $\alpha$ indicate that the distribution of clients' local datasets across classes is more uniform.

Table 2: Performance Comparison across various SOTA Algorithms on Various Datasets with 20% and 30% Non-IID Label Skew with low degree of quantity shift (Dirichlet parameter $\alpha' = 1$)

| Algorithm | 20% Label Skew | | | | 30% Label Skew | | | |
|---|---|---|---|---|---|---|---|---|
| | CIFAR-10 | FMNIST | SVHN | CIFAR-100 | CIFAR-10 | FMNIST | SVHN | CIFAR-100 |
| FedAvg | $46.20 \pm 0.97$ | $57.12 \pm 0.30$ | $74.61 \pm 0.36$ | $51.34 \pm 0.78$ | $57.28 \pm 0.17$ | $77.56 \pm 0.24$ | $68.34 \pm 0.45$ | $53.13 \pm 1.46$ |
| FedProx | $46.77 \pm 0.14$ | $56.81 \pm 0.16$ | $77.23 \pm 0.45$ | $53.38 \pm 0.86$ | $57.8 \pm 0.23$ | $73.87 \pm 0.25$ | $69.65 \pm 0.19$ | $53.97 \pm 0.85$ |
| PerFedAvg | $84.68 \pm 0.19$ | $91.18 \pm 0.21$ | $92.34 \pm 0.13$ | $69.43 \pm 0.22$ | $82.83 \pm 0.14$ | $94.74 \pm 0.17$ | $91.48 \pm 0.29$ | $60.70 \pm 0.30$ |
| Fedsoft | $77.42 \pm 0.21$ | $87.64 \pm 0.35$ | $90.48 \pm 0.24$ | $65.98 \pm 0.37$ | $76.94 \pm 0.38$ | $89.56 \pm 0.37$ | $84.86 \pm 0.45$ | $56.61 \pm 0.31$ |
| PACFL | $90.45 \pm 0.30$ | $94.41 \pm 0.31$ | $94.96 \pm 0.12$ | $70.35 \pm 0.36$ | $87.01 \pm 0.38$ | $97.28 \pm 0.24$ | $94.36 \pm 0.19$ | $63.91 \pm 0.76$ |
| CFL | $72.80 \pm 0.66$ | $86.97 \pm 0.23$ | $82.06 \pm 0.34$ | $61.43 \pm 0.92$ | $71.85 \pm 0.79$ | $85.67 \pm 0.23$ | $80.23 \pm 0.25$ | $52.90 \pm 1.17$ |
| IFCA | $89.68 \pm 0.17$ | $94.02 \pm 0.09$ | $93.28 \pm 0.13$ | $72.86 \pm 0.29$ | $86.42 \pm 0.25$ | $96.61 \pm 0.14$ | $92.86 \pm 0.19$ | $61.34 \pm 0.43$ |
| **FLAG** | $\mathbf{93.81 \pm 0.09}$ | $\mathbf{96.36 \pm 0.13}$ | $\mathbf{96.64 \pm 0.14}$ | $\mathbf{74.12 \pm 0.33}$ | $\mathbf{90.35 \pm 0.13}$ | $\mathbf{97.71 \pm 0.05}$ | $\mathbf{96.42 \pm 0.08}$ | $\mathbf{65.06 \pm 0.61}$ |

Table 3: Performance Comparison across various SOTA Algorithms on Various Datasets with 20% and 30% Non-IID Label Skew with high degree of quantity shift (Dirichlet parameter $\alpha' = 0.25$)

| Algorithm | 20% Label Skew | | | | 30% Label Skew | | | |
|---|---|---|---|---|---|---|---|---|
| | CIFAR-10 | FMNIST | SVHN | CIFAR-100 | CIFAR-10 | FMNIST | SVHN | CIFAR-100 |
| FedAvg | $42.02 \pm 1.17$ | $53.11 \pm 0.31$ | $69.79 \pm 0.51$ | $47.16 \pm 0.91$ | $54.24 \pm 0.08$ | $72.86 \pm 0.40$ | $64.15 \pm 0.64$ | $50.99 \pm 1.35$ |
| FedProx | $43.98 \pm 0.17$ | $53.61 \pm 0.20$ | $74.75 \pm 0.27$ | $50.56 \pm 0.70$ | $54.99 \pm 0.20$ | $68.22 \pm 0.16$ | $64.80 \pm 0.25$ | $48.66 \pm 0.80$ |
| PerFedAvg | $81.09 \pm 0.35$ | $86.51 \pm 0.19$ | $89.20 \pm 0.35$ | $65.59 \pm 0.02$ | $77.45 \pm 0.24$ | $89.77 \pm 0.15$ | $88.23 \pm 0.31$ | $57.38 \pm 0.10$ |
| Fedsoft | $76.44 \pm 0.18$ | $84.58 \pm 0.14$ | $83.75 \pm 0.33$ | $62.54 \pm 0.41$ | $72.48 \pm 0.17$ | $85.15 \pm 0.17$ | $82.43 \pm 0.40$ | $55.24 \pm 0.43$ |
| PACFL | $86.99 \pm 0.40$ | $91.90 \pm 0.47$ | $89.88 \pm 0.25$ | $66.11 \pm 0.29$ | $84.66 \pm 0.29$ | $91.96 \pm 0.25$ | $90.98 \pm 0.23$ | $58.30 \pm 0.56$ |
| CFL | $68.67 \pm 0.76$ | $81.90 \pm 0.10$ | $79.83 \pm 0.38$ | $57.38 \pm 0.95$ | $67.57 \pm 0.69$ | $80.64 \pm 0.21$ | $75.21 \pm 0.09$ | $49.63 \pm 1.29$ |
| IFCA | $86.64 \pm 0.13$ | $90.93 \pm 0.17$ | $89.51 \pm 0.10$ | $69.08 \pm 0.48$ | $83.45 \pm 0.37$ | $91.50 \pm 0.11$ | $88.81 \pm 0.09$ | $56.33 \pm 0.40$ |
| **FLAG** | $\mathbf{90.29 \pm 0.12}$ | $\mathbf{93.19 \pm 0.20}$ | $\mathbf{93.41 \pm 0.23}$ | $\mathbf{69.37 \pm 0.20}$ | $\mathbf{86.79 \pm 0.16}$ | $\mathbf{92.42 \pm 0.04}$ | $\mathbf{92.19 \pm 0.11}$ | $\mathbf{62.86 \pm 0.60}$ |

that clustering, the server uses the average validation score across a few clients as a cluster goodness metric. Figure 1 presents the average validation accuracy for each clustering, illustrating the relationship between different $\alpha$ values and the resulting number of clusters. For each dataset, we applied a class skew of 30%, combined with a low degree of quantity shift (Dirichlet parameter $\alpha'$ = 1), across four different datasets. In Figure 1, x-axis shows the $\alpha$ values and the y-axis shows the corresponding average validation score for that $\alpha$. The red line indicates the validation accuracy and blue bars represent the number of clusters at each value of $\alpha$. We can observe that $\alpha$=1 puts all the clients in a single cluster. As $\alpha$ decreases, more clusters are created, and we observe a steep increase in validation accuracy. This happens because more similar clients are being grouped into the same clusters, and disparate clients reside in separate clusters. Decreasing the $\alpha$ more at one point, halts the steep accuracy increase. At that point, the number of clusters has reached the underlying number of groups that have similar classes in the Non-IID distribution. The distribution which was formed as discussed above §5. Decreasing $\alpha$ further beyond this point and creating more clusters is detrimental to the FL training because it cannot benefit from the similar clients. Thus, we aim to keep the number of clusters minimal. We can use the elbow method (Syakur et al., 2018) to find the optimal point in $\alpha$. For example, in Figure 1(a), the optimal $\alpha$ using elbow method can be identified as $\alpha = 0.6$.

Figure 1: Comparison of average validation accuracy with cluster $\alpha$ values and number of clusters as a function of the distance threshold.

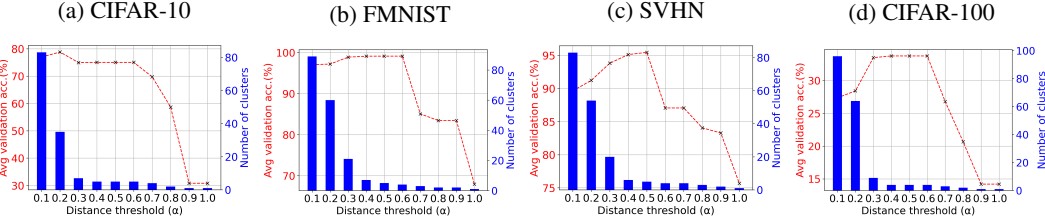

| (a) CIFAR-10 | (b) FMNIST | (c) SVHN | (d) CIFAR-100 |
|---|---|---|---|

## 5.3 ABLATION STUDY

We conducted ablation studies to see how combing gradient and data improves upon accuracy compared to just using either data or gradient alone. To achieve this, we set the combination ratio $\beta$=0 and $\beta$=1 to cluster clients with gradient only and data only similarity metric. Table 4 presents the accuracy metrics for the ablation studies and we also included the results of FedAvg as a baseline for comparison under the same distribution. The results clearly show that combining data and gradient significantly improves accuracy compared to using only the data or gradient similarity metric.

Table 4: Ablation study for effectiveness of combining gradient and data (denoted as G+D), compared to using just gradient (as G) and data (as D), non-IID, 20% and 30% label skew, $\alpha'$=1.

| Class skew | Dataset | FedAvg | G | D | G+D |
|---|---|---|---|---|---|
| 20% | CIFAR-10 | 46.20 ±0.97 | 87.47 ±0.64 | 89.95 ±0.13 | 93.81 ±0.09 |
| | SVHN | 74.61 ±0.36 | 84.82 ±0.14 | 94.91 ±0.13 | 96.64 ±0.14 |
| | FMNIST | 57.12 ±0.30 | 88.38 ±0.15 | 94.67 ±0.05 | 96.36 ±0.13 |
| | CIFAR-100 | 51.34 ±0.78 | 62.53 ±0.24 | 71.73 ±0.19 | 74.12 ±0.35 |
| 30% | CIFAR-10 | 57.48 ±0.17 | 81.34 ±0.52 | 87.93 ±0.09 | 90.31 ±0.13 |
| | SVHN | 68.34 ±0.45 | 90.05 ±0.16 | 94.29 ±0.13 | 96.42 ±0.08 |
| | FMNIST | 77.17 ±0.24 | 92.68 ±0.11 | 94.65 ±0.07 | 97.71 ±0.08 |
| | CIFAR-100 | 53.13 ±1.46 | 54.23 ±0.62 | 62.35 ±0.39 | 65.06 ±0.61 |

## 5.4 COMMUNICATION ROUND

In this section, we compare the performance of our proposed method with the rest of the SOTA under a limited communication budget of 80 rounds. We present the average final local test accuracy over all clients versus the number of communication rounds across four different datasets, with a Non-IID label skew (30%), in Figure 2. As we can see in Figure 2, FLAG takes between 20 to 30 communication rounds for the Non-IID label skew (30%) to reach convergence and reaches convergence faster than the other SOTA algorithms in all datasets.

Figure 2: Test accuracy versus number of communication rounds for Non-IID (30%), $\alpha'$=1.

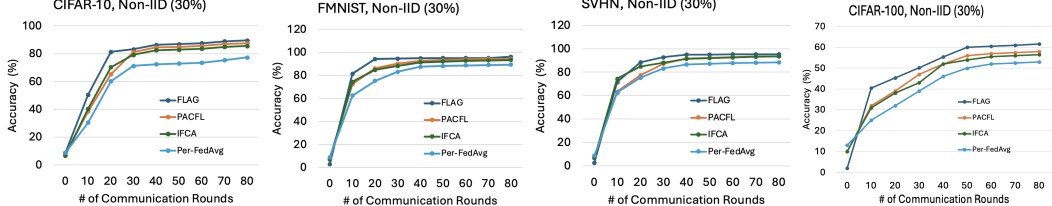

## 6 CONCLUSIONS

This work presents a novel algorithm, FLAG (Federated Learning with Data and Gradient), which addresses the limitations of existing clustered FL techniques and effectively tackles data heterogeneity challenges in FL. FLAG combines both data and gradient information to cluster clients more effectively, addressing a broader range of data heterogeneity issues. The algorithm leverages principal vectors and gradient similarity to create a robust proximity matrix, which is used for clustering via Agglomerative Hierarchical Clustering. Moreover, FLAG employs an efficient method to determine the optimal number of clusters, improving scalability and performance. Extensive experiments on various heterogeneous data distributions, including quantity shifts and class imbalances, demonstrate that FLAG outperforms existing approaches in terms of accuracy.

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

# 7 APPENDIX

# 8 ALGORITHM TECHNICAL ISSUES

In this section we will discuss the additional technical issues related to the FLAG algorithm.

**Problem formulation** FLAG, which is a cluster FL problem, can be formulated as a standard empirical risk minimization (ERM) task. The objective is to learn parametric models by minimizing a loss function defined over the data. We assume the presence of one server and $N$ clients. The server and clients communicate using a predefined communication protocol. Additionally, we consider $Z$ distinct data distributions, $\rho_1, \ldots, \rho_Z$, with the $N$ clients partitioned into $Z$ disjoint clusters, $\mathbb{C}_1, \ldots, \mathbb{C}_Z$. We assume that the cluster identities of the clients are not revealed to FLAG. Each client $i \in \mathbb{C}_z$ holds a dataset $D_i$, which may be non-iid and subject to various types of data skews. The goal is to minimize the loss function $F_z(\theta) := \mathbb{E}_{i \sim D_z}[f(\theta; d)]$ for all $z \in \{1, \ldots, Z\}$, where $f(\theta; d)$ represents the loss associated with a data point $d \in D_i$ for client $i \in \{1, \ldots, N\}$.

**Privacy constraints** Privacy is an important factor of FL, as it aims to enable collaborative model training while concealing the sensitive data of individual clients. In this section we will discuss the privacy constraints and concerns of our proposed FLAG. Regarding weighted class-wise data-based similarity detection, clients share a small set of principal vectors with the server. Additionally, each client's class frequency information is shared with the server to compute the weights for similarity measurements. However, in FLAG, the privacy of the clients' data is still preserved since the principal vectors are linear combinations of the data points and not the actual data themselves. Moreover, the number of principal vectors shared with the server is less than $1\%$ of the size of the dataset for each class per client. This approach aligns with prior works, such as Vahidian et al. (2023). In privacy-sensitive scenarios, additional privacy mechanisms such as those proposed by Bonawitz et al. (2017), encryption methods, or differential privacy techniques can be used to prevent information leakage and provide stronger protection. Similarly, these privacy-preserving methods can be employed when sharing class frequency information. Alternatively, in more privacy sensitive cases, uniform weights can be used for similarity measurements, eliminating the need to share class

frequency information entirely. Additionally, to address concerns regarding information leakage in gradient-based similarity detection,

**Time complexity of FLAG** The computational complexity of the federated learning part of Flag (Algorithm 1, line 12-19) is largely comparable to the other clustered federated learning (FL) approaches. Algorithm 3 is essentially iterating over different clusterings to find the best cluster formation. We tried to reduce time required to do so in Algorithm 3 compared to other literature. We adopted a framework that uses a lightweight model with fewer layers, employing a fraction of the clients, running fewer rounds of training. The clustering part of the Flag which can be denoted by (Algorithm 1, line 6-11), which also incorporates Algorithm 2, is the main source of additional computational overhead compared to vanilla FedAvg (McMahan et al., 2017). That said, our approach uses a one-shot clustering strategy, meaning clustering is performed only once at the start to assign each client a fixed cluster ID. This is more efficient than other methods (e.g., Ghosh et al. (2020), Ruan & Joe-Wong (2022)) that perform iterative clustering, where client clusters are repeatedly updated at every training round until the FL process ends. Compared to the aforementioned approaches our approach is much more efficient since we are performing the clustering one time and obtaining the final cluster ID's.

In this part, we analyze the overhead of the clustering part of the Flag. As referred to Algorithm 1, line 8, each client performs SGD training on its local data before sending the local update to the server. Here, each client performs at least 2 rounds of SGD training with local epochs of 10 steps for each client. As for applying SVD to extract principal vectors, referred to in (Algorithm 1, line 8), each client applies SVD on the dataset of each class. Which takes less time compared to another approach Vahidian et al. (2023) that performs SVD on each client's whole dataset. It is because performing SVD on a large dataset of a client with $N$ data points, $F$ features, and assuming $N > F$, would cost $\mathcal{O}(FN^2)$. But, partitioning the dataset into $C$ classes where each partition is of size approximately $N/C$, the total computational cost becomes: $C \cdot \mathcal{O}\left(F \cdot \left(\frac{N}{C}\right)^2\right) = \mathcal{O}\left(\frac{FN^2}{C}\right)$. Here, SVD benefits from the quadratic relationship with the number of data points. This is a reduction by a factor of $C$ compared to performing SVD on the entire dataset of size $N$.

**The communication complexity** The FL part of Flag has communication complexity that is similar to other notable cluster FL approaches (Vahidian et al., 2023). Since, Flag is a one-shot clustering, during the FL training FLAG doesn't do any clustering. Which is faster compared to some of the other iterative clustering approaches (Ghosh et al., 2020; Sattler et al., 2020). Other sources of communication burden relies on building the proximity matrix. Prior to start of the FL protocol, each client, after running about 20 epochs of local training, sends the gradient update to the server. This is the main overhead, but I also want to point out that this is done only one time with one-shot clustering. Compared to some of the other approaches that on each iteration sends gradient update or use gradient information to update cluster identities. Another part of computing the proximity matrix is sending the principal angles to the server by each client. But, the communication burden is very small here because the number of principal vectors shared with the server is less than 1% of the size of the dataset for each class per client. Since, the data was very non-IID with each client having arbitrary number of classes, the overall accuracy was lower compared to the previous data partition. But, FLAG was robust in capturing similar clients yielding huge accuracy advantages over other algorithms.

## 9 EXPERIMENT ON ADDITIONAL DATA PARTITION

We conducted experiments on an additional data partition to investigate the robustness of our FLAG algorithm. This particular data distribution is achieved by sampling client data using a Dirichlet distribution over categories (Hsu et al., 2019; Wu et al., 2022). The process can be summarized as follows: each client samples data based on probability factors derived by multiplying the Dirichlet concentration factor ($\alpha'$) with the relative label popularity. This yields a multinomial distribution over labels, controlling the data heterogeneity among clients. As $\alpha'$ approaches 0, each client receives data from a single category, whereas as $\alpha'$ approaches infinity, the client distribution mirrors the overall label popularity. Table 5 presents the performance comparison of different algorithms under this data partition, where the Dirichlet concentration factor $\alpha'$ is set to 0.25. We set a lower value for $\alpha'$ to achieve a more heterogeneous dataset. The hyperparameters of the experiment are

as follows: $\delta = 0.5$, $t_g = 3$ rounds of SGD training on local data, each client performing 10 local epochs, $m' = 25$ sampled clients for optimal clustering, and an interval size of $\alpha = 0.01$.

Table 5: Performance Comparison across various SOTA Algorithms on Various Datasets with Non-IID data based on Dirichlet distribution over categories, $\alpha' = 0.25$

| Algorithm | CIFAR-10 | FMNIST | SVHN |
|---|---|---|---|
| FedAvg | 55.32±0.19 | 79.52±0.24 | 72.41±0.18 |
| Fedsoft | 53.54±0.31 | 82.24±0.29 | 79.81±0.25 |
| PACFL | 59.88±0.22 | 87.12±0.14 | 84.42±0.44 |
| CFL | 42.84±0.13 | 81.85±0.31 | 80.43±0.14 |
| IFCA | 57.21±0.27 | 85.23±0.41 | 83.75±0.52 |
| **FLAG** | **64.34 ± 0.18** | **89.83±0.13** | **87.14 ±0.23** |

## 10 HYPERPARAMETERS TUNING

In the context FLAG, hyperparameters play a crucial role in determining the model's performance, stability, and robustness. To better understand the effectiveness of FLAG, we investigate how sensitive the algorithm is to variations in different hyperparameters.

**The combination ratio** $\beta$ represents the weighting ratio between the data adjacency matrix and the gradient adjacency matrix during their aggregation to compute the final proximity matrix. Table 6 presents the accuracy metrics for various values of $\beta$, ranging from 0 to 1 in increments of 0.1. The additional hyper-parameter settings for the experiment are as follows: $\rho = 20\%$, Dirichlet $\alpha' = 1$, $\delta = 0.5$, $t_g = 3$ rounds of SGD training on local data, each client performing 10 local epochs, $m' = 25$ sampled clients for optimal clustering, and an interval size of $\alpha = 0.1$.

Higher $\beta$ values signify greater emphasis on the data adjacency matrix, whereas lower $\beta$ values prioritize the gradient adjacency matrix. From Table 6, we observe that in most cases, the data-only adjacency matrix ($\beta = 1$) yields accuracy values closer to the peak, while the gradient-only matrix ($\beta = 0$) results in lower accuracy comparatively. As $\beta$ approaches the middle range, where both data and gradient matrices receive roughly equal emphasis, FLAG exhibits higher accuracy by achieving more robust clustering. Additionally, around this middle range of $\beta$ values, FLAG tends to produce nearly identical clustering, leading to similar accuracy across a range of $\beta$ values. This phenomenon can be attributed to our data distribution, where clients can be grouped into disjoint sets based on their data distributions. As a result, FLAG produces consistent clustering even when $\beta$ values vary slightly. Depending on the specific data distributions, $\beta$ can be adjusted slightly (e.g., currently set at $\beta = 0.5$) to fine-tune the algorithm and evaluate whether it yields different results.

**Weight range of data similarity matrix** $\delta$ is the parameter to determine the importance of weights for computing the data similarity matrix. Since a small set of principal vectors is used to determine the similarity between clients, the weights take into account the size of those compared datasets. The weighting scheme ensures the similarity decreases if there are significant differences in the dataset size. The degree of how much it will decrease is determined by the $\delta$ parameter. Smaller values of $\delta$ determine the impact of weights would be minimal and higher values of $\delta$ denote a more significant impact. After computing weights for each data similarity value, the weights are normalized within a given range of $[1 - \delta, 1 + \delta]$. We consider a range of [0,1) for $\delta$, which denotes the similarity value/cosine angle can increase by a factor of $\approx$2 at most. Table 7 demonstrates how the accuracy metric changes over different values of $\delta$ on different datasets. Since the weighting focuses on size difference/quantity shift, we run our experiments on different ranges of quantity shifts, which is denoted in Table 7 with $\alpha'$=1 and 0.25. As you can see from the table, for $\alpha'$ the accuracy pretty much stays the same regardless of what $\delta$ value is used. This happens since the distribution contains less quantity shift with smaller differences in class values. For $\alpha'$=0.25 we can observe a slight increase in accuracy for some cases. Since, in this case, the difference in class data size can lead to adjusted similar values and different clustering.

**Local steps** $t_g$ determines the number of local training steps each client performs on its local data before sending the local gradient direction to the server. Since clients undergo multiple epochs of local training before sending the local update to the server, their local models are already partially converged with respect to the local data. The server uses the gradient directions to compute the co-

Table 6: Accuracy metrics for various values of $\beta$ (ranging from 0 to 1 in increments of 0.1), non-IID, with 20%, Dirichlet $\alpha' = 1$.

| Class Skew | $\beta$ | CIFAR-10 | SVHN | FMNIST |
|---|---|---|---|---|
| | 0.0 | 87.47±0.64 | 84.82±0.14 | 88.38±0.24 |
| | 0.1 | 89.79±0.28 | 88.02±0.16 | 91.44±0.21 |
| | 0.2 | 91.88±0.21 | 91.35±0.23 | 93.72±0.19 |
| | 0.3 | 92.33±0.29 | 94.79±0.22 | 95.34±0.31 |
| | 0.4 | 93.81±0.09 | 96.64±0.14 | 96.36±0.13 |
| 20% | 0.5 | 93.81±0.09 | 96.64±0.14 | 96.36±0.13 |
| | 0.6 | 93.81±0.09 | 96.64±0.14 | 96.36±0.13 |
| | 0.7 | 93.81±0.09 | 96.64±0.14 | 96.36±0.13 |
| | 0.8 | 93.81±0.09 | 96.31±0.28 | 96.36±0.13 |
| | 0.9 | 91.12±0.23 | 96.31±0.28 | 95.21±0.08 |
| | 1.0 | 89.95±0.13 | 94.91±0.13 | 94.67±0.05 |
| | 0.0 | 81.34±0.52 | 90.05±0.16 | 92.68±0.11 |
| | 0.1 | 83.83±0.39 | 92.01±0.19 | 94.02±0.14 |
| | 0.2 | 85.87±0.32 | 94.03±0.43 | 95.67±0.10 |
| | 0.3 | 88.48±0.19 | 96.42±0.08 | 96.14±0.11 |
| | 0.4 | 90.31±0.23 | 96.42±0.08 | 97.71±0.08 |
| 30% | 0.5 | 90.31±0.23 | 96.42±0.08 | 97.71±0.08 |
| | 0.6 | 90.31±0.23 | 96.42±0.08 | 97.71±0.08 |
| | 0.7 | 90.31±0.23 | 96.42±0.08 | 97.71±0.08 |
| | 0.8 | 89.68±0.24 | 96.42±0.08 | 96.48±0.09 |
| | 0.9 | 88.12±0.21 | 95.57±0.24 | 95.21±0.17 |
| | 1.0 | 87.47±0.13 | 94.29±0.13 | 94.65±0.07 |

Table 7: Accuracy metrics for various values of $\delta$, non-IID, with 20% class, Dirichlet $\alpha' = 1$ and $\alpha' = 0.25$.

| Quantity shift | $\delta$ | CIFAR-10 | SVHN | FMNIST |
|---|---|---|---|---|
| | 0.0 | 93.38±0.21 | 96.64±0.14 | 95.84±0.13 |
| | 0.2 | 93.38±0.21 | 96.64±0.14 | 96.36±0.13 |
| | 0.4 | 93.81±0.09 | 96.64±0.14 | 96.36±0.13 |
| | 0.6 | 93.81±0.09 | 96.64±0.14 | 96.36±0.13 |
| | 0.8 | 93.81±0.09 | 96.64±0.14 | 96.36±0.13 |
| $\alpha' = 1$ | 1.0 | 93.81±0.09 | 96.64±0.14 | 96.36±0.13 |
| | 0.0 | 88.85±0.25 | 92.42±0.08 | 91.97±0.30 |
| | 0.2 | 89.32±0.32 | 92.42±0.08 | 92.67±0.10 |
| | 0.4 | 89.32±0.32 | 93.19±0.23 | 93.41±0.23 |
| | 0.6 | 90.29±0.12 | 93.19±0.23 | 93.41±0.23 |
| | 0.8 | 90.29±0.12 | 93.19±0.23 | 93.41±0.23 |
| $\alpha' = 0.25$ | 1.0 | 90.29±0.12 | 93.19±0.23 | 93.41±0.23 |

sine similarity between clients and form the gradient similarity matrix. The number of local steps $t_g$ is important as it impacts the trade-off between efficiency and accuracy. We want the number of local epochs to be as small as possible while also maintaining the convergence of gradient so the similarity between clients can be properly determined. Table 8 demonstrates how the accuracy metric changes over different values of $t_g$ on different datasets. To properly exhibit the effects of $t_g$, without the influence of the data similarity matrix, we set the $\beta=0$ to tune out the data matrix. We perform the experiment on a dataset with class skew and quantity shift combined as $\rho = 20\%$, Dirichlet $\alpha' = 1$, $m' = 25$ sampled clients for optimal clustering, and an interval size of $\alpha = 0.1$. As shown in the table, increasing $t_g$ increases the accuracy of the models, since the gradients become more converged and similar clients show similar gradient updates. After reaching around $t_g=20$ the accuracy halts in most cases. This indicates at that point the gradient convergence and gradient update direction have already established, and training local models any more will not produce any different similarity values.

Table 8: Accuracy metrics for various values of $t_g$ over various dataset, non-IID, with 20% class, Dirichlet $\alpha' = 1$.

| Combination Ratio | $t_g$ | CIFAR-10 | SVHN | FMNIST |
|---|---|---|---|---|
| $\beta = 0$ | 0 | 46.20±0.97 | 74.61±0.36 | 57.12±0.30 |
| | 5 | 68.54±0.72 | 79.44±0.68 | 76.86±0.23 |
| | 10 | 77.81±0.51 | 82.32±0.41 | 83.51±0.49 |
| | 15 | 84.81±0.59 | 84.82±0.24 | 88.38±0.15 |
| | 20 | 87.47±0.64 | 84.82±0.24 | 88.38±0.15 |
| | 25 | 87.47±0.64 | 84.82±0.24 | 88.38±0.15 |

