# OpenReview forum: "FLAG: Clustered Federated Learning Combining Data and Gradient Information in Heterogeneous Settings"
_ICLR.cc/2025/Conference — Submitted to ICLR 2025_

### Official Review · Reviewer_fBmA · 2024-10-30

**Soundness:** 2
**Presentation:** 2
**Contribution:** 2
**Rating:** 5
**Confidence:** 4

**Summary:**

In this work, the authors propose FLAG, a clustered-based federated learning (FL) method that integrates both data similarity and gradient similarity to effectively cluster clients, addressing challenges posed by heterogeneous data distributions. FLAG also employs optimal clustering to perform hierarchical clustering efficiently, enabling a systematic search for the ideal number of clusters, which ultimately enhances model performance in diverse FL settings.

**Strengths:**

1 The authors introduce a weighted class-wise similarity metric combining data and gradient information, which enhances clustering accuracy and robustness to different data skews.

2 FLAG has better performance among different experiments, and provides faster convergence for FL.

3. The motivation of clustering clients with different data heterogeneity is interesting and useful.

**Weaknesses:**

1 The methodology takes one-shot clustering, which might be inconsistent for gradient similarity during the iterative federated process.

2 The authors can enhance the discussion and survey about related works, e.g., FedAC[1] and FedRC[2].

3 It lacks deeper theoretical analysis from three aspects, (1) privacy leakage of sharing both data and gradient knowledge to server, (2) the communication convergence based on hierarchical clustering, and (3) the communication and  computation burden for FLAG.

4 The experiments are not in accordance with the motivation of FLAG. (1) some data heterogeneity types are not evaluated, e.g., concept drift and concept shift. (2) The vital hyper-parameter sensitivities are overlooked, e..g., the sample ratio of clients, and the total number of clients. (3) some crucially related works, e.g., FedAC[1] and FedRC[2].

[1] Zhang Y, Chen H, Lin Z, et al. FedAC: A Adaptive Clustered Federated Learning Framework for Heterogeneous Data[J]. arXiv preprint arXiv:2403.16460, 2024.
[2] Guo Y, Tang X, Lin T. FedRC: Tackling Diverse Distribution Shifts Challenge in Federated Learning by Robust Clustering[C]//Forty-first International Conference on Machine Learning.

**Questions:**

Q1: Can FLAG adapt to other data modality?

Q2: Can authors explain the necessity of using hierarchical clustering?

Q3: Can authors explain difference between determining the number of clusters and setting a sequence of thresholds  for cluters? It seems that they have the same effect for clustering.

**Details Of Ethics Concerns:**

The approach publish both data and gredient knowledge to server, bringing the privacy risk.

---

### Official Review · Reviewer_r5EU · 2024-10-31

**Soundness:** 2
**Presentation:** 3
**Contribution:** 2
**Rating:** 6
**Confidence:** 4

**Summary:**

This paper presents a clustered FL method named FLAG, which takes both raw data similarity and gradients similarity into account to accurately group clients. Experiments conducted on multiple datasets and models demonstrate the superior performance of FLAG.

**Strengths:**

1. The method described in the paper is clear and intuitive, and the readability of the paper is good.
2. The background part is written well, and the problem the authors aim to address is clear.
3. Experiments conducted in the paper effectively show the superior performance of FLAG compared to competing methods.

**Weaknesses:**

1. The authors do not discuss whether FLAG can prevent privacy leakage, a critical concern in federated learning (FL). Specifically, they propose using truncated Singular Value Decomposition (SVD) to decompose raw data into singular values, which, along with gradients, are then sent to the server for clustering. However, previous research [1] has demonstrated that gradients can be exploited in gradient leakage attacks to reveal clients' local data. While this paper is not primarily focused on security, my concern is that the inclusion of singular values may inadvertently facilitate such attacks. Could the authors discuss the privacy implications of FLAG, particularly regarding the transmission of singular values and gradients? Also, the authors could compare the privacy risks of FLAG to existing methods or propose potential mitigations.
2. The paper does not provide analyses of the time and space complexity for the three algorithms. Additionally, truncated SVD is known to be challenging for parallel processing, and FLAG necessitates running truncated SVD on every class of data, which could impact the practical applicability of FLAG. Could the authors provide time and space complexity analyses for three algorithms, and to discuss how the use of truncated SVD on every class might affect scalability and practical implementation?
3. FLAG performs one-shot clustering during the first iteration. Previous research has demonstrated that, during training, clients' gradients are nearly aligned (i.e., point in the same direction) in the early stages and diverge later. Despite this, Table 4 indicates that one-shot clustering based solely on gradients in the first iteration is effective. Could the authors provide similarity heat maps of G, D and G+D in different iterations during training or explain why using only gradients in the first iteration is sufficient for clustering?
4. The authors claim that the "predefined cluster numbers" is a limitation of existing clustered FL methods. However, the proposed FLAG method also needs to search for an optimal clustering threshold to achieve the best performance. In the paper, the elbow method is used for determining this threshold, similar to how many current clustered FL methods [2] determine the optimal cluster numbers. Therefore, it is unclear how FLAG effectively addresses the "predefined cluster numbers" limitation. Could the authors clarify how the approach to determining the optimal clustering threshold in FLAG differs from or improves upon existing methods for determining cluster numbers?
5. cluster FL -> clustered FL?

I will surely increase my rating if my concerns are well addressed.

[1] Zhu, L., Liu, Z., & Han, S. (2019). Deep leakage from gradients. Advances in neural information processing systems, 32.

[2] Zhou, Y., Shi, M., Tian, Y., Li, Y., Ye, Q., & Lv, J. (2024, April). Federated CINN Clustering for Accurate Clustered Federated Learning. In ICASSP 2024-2024 IEEE International Conference on Acoustics, Speech and Signal Processing (ICASSP) (pp. 5590-5594). IEEE.

**Questions:**

Please see weaknesses above.

---

### Official Review · Reviewer_VVcu · 2024-11-03

**Soundness:** 1
**Presentation:** 2
**Contribution:** 1
**Rating:** 1
**Confidence:** 4

**Summary:**

The paper introduces FLAG, a clustered federated learning (FL) method that combines both data and gradient similarities to group clients in highly heterogeneous settings. FLAG aims to enhance client clustering accuracy by using a weighted similarity metric, incorporating principal vectors from data and cosine angles between gradients.

**Strengths:**

The paper studies clustered FL problem and its challenges which is an important topic.

**Weaknesses:**

The paper presents an approach for clustered federated learning (FL) with data and gradient similarity. While this is an interesting concept, I have several concerns. See my comments below:

See my comments below
* The abstract and introduction mention addressing limitations of prior clustered FL methods under complex data heterogeneity scenarios, such as concept shift and drift. However, these claims are not supported by experiments or analyses specifically targeting these scenarios.
* The paper contains several statements that are incorrect or need justifications: 1) IFCA is inaccurately characterized as gradient-based clustering. 2) The claim that using only gradient or data similarity independently is insufficient (lines 78-81) requires justification. In particular, gradient similarity may not capture underlying data or task similarity if clients have different model weights or objectives. 3) The explanation of PACFL’s approach (lines 88-91) does not align with its actual methodology, which measures subspace angles to assess data similarity.
* Table 1 presents performance comparisons without sufficient context on the FL setup, making it challenging to interpret the results.
* The discussion on related work in clustered FL lacks depth and omits several state-of-the-art approaches. A comprehensive literature review section, addressing the broader landscape of clustered FL methods is required.
* The paper does not clarify how gradients are communicated within the FL framework. If the approach involves FedSGD, it would be helpful to specify whether gradients or local updates are exchanged in each communication round.
* The proposed algorithm for determining the optimal clustering is a brute-force approach and is unlikely to be communication-efficient. Furthermore, the algorithm lacks any novel aspect.
* The HC clustering threshold depends on the similarity matrix values (lines 280-281). It is important to clarify whether the similarity matrix is normalized, as this impacts the thresholding and clustering results.
* The experimental section overall is weak and does not provide details about the actual FL setup. For example, in Table 2, it is unclear whether different data heterogeneity settings are mixed.

**Questions:**

See my comments above.

---

> ### Comment · Reviewer_VVcu · 2024-11-27
> **Response to the authors**
>
> Thank you for your response. However, I believe that the paper needs significant revisions and seems it is not yet complete as pointed out by the authors. Furthermore, I believe that gradient similarity cannot capture any correct information regarding the underlying data on non-convex NNs, specially when the clients are taking multiple gradient updates and no longer have same weights after the first update. I recommend the authors to rigorously investigate the technical correctness of claims and provide proof. Therefore, I keep my score.

---

### Official Review · Reviewer_q22U · 2024-11-04

**Soundness:** 2
**Presentation:** 2
**Contribution:** 2
**Rating:** 5
**Confidence:** 5

**Summary:**

FLAG is a method for clustered federated learning (FL), which describes a family of FL methods that tackle data heterogeneity across clients by clustering clients in groups and training separate global models for each group. Unlike previous approaches that rely on either data similarity or gradient similarity to compose clusters, FLAG integrates both pieces of information for an initial one-shot clustering of the client population. The groups defined in that initialization phase are then maintained fixed throughout the cluster FL step to obtain the global models for each cluster. FLAG brings performance improvements spanning from 0.5% to 4% compared to the chosen three baselines. FLAG loses only one comparison across four image datasets, each of which is partitioned using two custom methods.

**Strengths:**

The authors propose a promising and effective alternative to previous proposals for tackling the challenges of the clustered FL setting. The method FLAG proposed in this work is well motivated by the research gap in the literature and is the first to combine both data and gradient similarity approaches to cluster clients. The authors demonstrate that they have critically familiarized themselves with the literature on clustered FL and its related topics.

**Weaknesses:**

I thank the authors for submitting this interesting work proposing the new FLAG method. Despite this work having some promising aspects, I found a number of weaknesses that, I believe, could be easily addressed to improve the paper. I hope the authors will critically consider these and find them useful for improving their work.

W1. Despite FLAG combining a novel approach, the algorithms used by the entire method are not novel, so FLAG can be considered an incremental contribution.

W2. The federated learning (FL) setting is not fully characterized, making it difficult to locate this work in the literature and to assess its impact and potential limitations. Describing fully the federated setting, with assumptions and examples, could give the readers more grounds to understand FLAG. Also, it might suggest to the FLAG’s authors means of improving evaluation.

W3. The method introduces several hyperparameters that are notably challenging to tune in FL. The experimental section doesn’t fully address practitioners' concerns about the sensitivity of FLAG’s performance to these hyperparameters. $\beta$, introduced at line 276, is the only one that is investigated even though it’s in the context of the ablation study for gradient and data-based proximity matrix.  Hyperparameters such as $\delta$ (line 243), number of local steps at line 261 (Alg.1 - line8), the interval size for the $\alpha$ at line 281 (Alg.3 - line 6), the number of cluster FL rounds $t\prime$ (Alg.3 - line 9), number of sampled clients to obtain the optimal clustering $m\prime$ (Alg.3 - line 2). (I am sorry to point out both algorithms and text, but, unfortunately, there is no complete overlap)

W4. At the intersection between the FL setting and the method, this work fails to address the privacy concerns relative to sharing the server information regarding the local data distribution. The authors must discuss the privacy implications of their method, particularly the one-shot clustering step, which involves sharing information strongly related to the clients' private data.

W5. At the intersection between the FL setting and the method, this work fails to address the scalability concerns of FLAG when applied to large-scale federated populations. The authors must discuss the scalability properties of their method and potential downsides related to trading-off between ML efficiency and scalability, particularly for the one-shot clustering step.

W6. Despite an extensive literature review in the Introduction (Sec.1), a complete and detailed comparison between the algorithms used by FLAG and the proposals in previous works, such as PACFL [1], CFL [2], and IFCA [3], deserves a more explicit and evident discussion.

W7. The experimental setting appears to rely extensively on the partitioning methodology used to obtain federated client datasets from a “centralized” dataset. Unfortunately, it seems to be chosen specifically for this method, and the literature does not sufficiently support it. A reader may wonder what the results would be when adopting other, more standard partitioning approaches.

W8. Some sentences throughout the paper, particularly in the evaluation section, sound very speculative and should be modified by strong theoretical/empirical support to justify them.

W9. The evaluations section appears very limited in length instead of the very long and verbose “Introduction” and central section (Section 3). Supporting the claims of this work necessitates a more extensive evaluation. In particular, some of the claims are not well supported or explained, such as “efficiency” in third contribution in introduction, “optimality” of the clustering obtained (is it empirical or theoretical?), “capturing data heterogeneity” in second contribution in introduction, “scalability improvements” in conclusions,  or just speculative, such as lines 420-422, lines 271-272 (clustering inaccuracies), lines 274-275 (combining leads to more accurate clusters),

W10. Most researchers nowadays tend to separate “Introduction” from “Related Works” and “Background”. This helps a lot with the narrative flow and conveys the paper's message. I strongly recommend to modify Section 1 to reflect such practices. As it stands, the narrative breaks several times in the introduction, which feels excessively long. “Related Works” can be put before Conclusions.

W11. Similar to the above, the abstract seems to be verbose in introducing the problem and describing the research gap. As a result, the number of lines reserved for the paper’s proposal is very minimal. Also, I suggest adding some numerical results to the abstract to convey the impact of FLAG more directly.

W12. I couldn’t help noticing that the baselines used in this work are relatively old. Comparing with the latest works would strengthen the evaluation and impact of this paper.

 [1] Saeed Vahidian, Mahdi Morafah, Weijia Wang, Vyacheslav Kungurtsev, Chen Chen, Mubarak Shah, and Bill Lin. Efficient distribution similarity identification in clustered federated learning via principal angles between client data subspaces. In Proceedings of the AAAI conference on artificial intelligence, volume 37, pp. 10043–10052, 2023.

 [2] Felix Sattler, Klaus-Robert Muller, and Wojciech Samek. Clustered federated learning: Model-agnostic distributed multitask optimization under privacy constraints. IEEE transactions on neural networks and learning systems, 32(8):3710–3722, 2020.

 [3] Avishek Ghosh, Jichan Chung, Dong Yin, and Kannan Ramchandran. An efficient framework for clustered federated learning. Advances in Neural Information Processing Systems, 33:19586–19597, 2020.

**Questions:**

I thank the authors for submitting an interesting work discussing a method for clustered federated learning. As a premise to the following questions, I declare that I would be very happy to increase the score if my concerns are fully addressed as the proposed work seems promising, achieving satisfactory results.

Q1. Can the authors provide additional experiments investigating the sensitivity/robustness of FLAG to the following hyperparameters? Hyperparameters such as $\beta$ (introduced at line 276), $\delta$ (line 243), number of local steps at line 261 (Alg.1 - line8), the interval size for the $\alpha$ at line 281 (Alg.3 - line 6), the number of cluster FL rounds $t^\prime$ (Alg.3 - line 9), number of sampled clients to obtain the optimal clustering $m^\prime$ (Alg.3 - line 2). For the parameter $\beta$, which has been used for the ablation study, it would be interesting to see more values and not just the extremes of its domain.

Q2. Can the authors discuss in detail what the federated learning setting used in this work looks like? I strongly recommend using [1] as a guideline to inspire from. I am particularly interested in the privacy aspects related to the federated setting and its scale. This must be put in the context of the proposed FLAG method. Such a discussion is usually mandatory in FL papers as it also helps draw comparisons with the literature, which would make the paper stronger. Real-world examples that reflect on the assumptions made will certainly help.

Q3. Can the authors extend the evaluation by including a new set of partitioning methods, such as some from [2]? Additionally, I would like to see a mini-benchmark of the clustering capability of FLAG. This could be carried out by modeling a client population with ground-truth clustering membership, either based on the labels of their samples (I suggest partitioning as it’s done for Cluster CIFAR100 dataset in [3]) or their feature (partitioning SVHN based on its data features is a reasonable option). I believe that such an experiment can make the clustering capabilities of FALG more straightforward to show, improving the paper.

Q4. Can the authors give more justification, formal demonstrations, references, context, or empirical observations to motivate the following claims? lines 024-026, show that past methods fail to capture the full spectrum of data heterogeneity; lines 071-072: limited flexibility; lines 079-082; line 088: “producing incorrect similarity”; lines 271-272; lines 274-275; lines 420-422; lines 461-463; line 527: scalability improvement; line 525; “broader range” compared to what?

Q5. Can you provide a detailed description that compares step-by-step the differences between FALG and the previous proposal? Even though there aren’t previous works combining both data and gradient similarity, I believe there’s a lot of value in detailing the data-based similarity of FLAG with previous data-based similarity clustering methods [4] and the gradient-based similarity part of FLAG with previous gradient-based similarity methods [5,6]. This discussion will highlight FALG’s contributions and improve the paper.

Q6. Can the authors compare their proposal with the following works [7,8,9], discussing both the method’s differences and the empirical results? The addition of these recent baselines will add a lot of value to this work.

Q7. There are some typos and grammar errors here and there. Some examples follow. line 016: i.i.d. contains the word “distributed”, so “each client’s data is distributed i.i.d.” is grammatically wrong; lines 128-131; line 170 is tautological; in table 4, the value of $\alpha^\prime$ is 1, and there are some mistakes in reporting numbers; in figure 2, don’t repeat the name of the dataset twice and improve the formatting.

 [1] Jianyu Wang, Zachary Charles, Zheng Xu, Gauri Joshi, H. Brendan McMahan, Blaise Agüera y Arcas, Maruan Al-Shedivat, Galen Andrew, Salman Avestimehr, Katharine Daly, Deepesh Data, Suhas N. Diggavi, Hubert Eichner, Advait Gadhikar, Zachary Garrett, Antonious M. Girgis, Filip Hanzely, Andrew Hard, Chaoyang He, Samuel Horváth, Zhouyuan Huo, Alex Ingerman, Martin Jaggi, Tara Javidi, Peter Kairouz, Satyen Kale, Sai Praneeth Karimireddy, Jakub Konečny, Sanmi Koyejo, Tian Li, Luyang Liu, Mehryar Mohri, Hang Qi, Sashank J. Reddi, Peter Richtárik, Karan Singhal, Virginia Smith, Mahdi Soltanolkotabi, Weikang Song, Ananda Theertha Suresh, Sebastian U. Stich, Ameet Talwalkar, Hongyi Wang, Blake E. Woodworth, Shanshan Wu, Felix X. Yu, Honglin Yuan, Manzil Zaheer, Mi Zhang, Tong Zhang, Chunxiang Zheng, Chen Zhu, & Wennan Zhu (2021). A Field Guide to Federated Optimization*. CoRR, abs/2107.06917.*

 [2] Shanshan Wu, Tian Li, Zachary Charles, Yu Xiao, Ken Liu, Zheng Xu, & Virginia Smith (2022). Motley: Benchmarking Heterogeneity and Personalization in Federated Learning. In *Workshop on Federated Learning: Recent Advances and New Challenges (in Conjunction with NeurIPS 2022)*.

 [3] Aviv Shamsian, Aviv Navon, Ethan Fetaya, and Gal Chechik. Personalized federated learning using hypernetworks. In Marina Meila and Tong Zhang, editors, Proceedings of the 38th International Conference on Machine Learning, ICML 2021, 18-24 July 2021, Virtual Event, volume 139 of Proceedings of Machine Learning Research, pages 9489–9502. PMLR, 2021. URL http://proceedings.mlr.press/v139/shamsian21a.html.

 [4] Saeed Vahidian, Mahdi Morafah, Weijia Wang, Vyacheslav Kungurtsev, Chen Chen, Mubarak Shah, and Bill Lin. Efficient distribution similarity identification in clustered federated learning via principal angles between client data subspaces. In Proceedings of the AAAI conference on artificial intelligence, volume 37, pp. 10043–10052, 2023.

 [5] Felix Sattler, Klaus-Robert Muller, and Wojciech Samek. Clustered federated learning: Model-agnostic distributed multitask optimization under privacy constraints. IEEE transactions on neural networks and learning systems, 32(8):3710–3722, 2020.

 [6] Avishek Ghosh, Jichan Chung, Dong Yin, and Kannan Ramchandran. An efficient framework for clustered federated learning. Advances in Neural Information Processing Systems, 33:19586–19597, 2020.

 [7] Y. Yan, X. Tong and S. Wang, "Clustered Federated Learning in Heterogeneous Environment," in IEEE Transactions on Neural Networks and Learning Systems, vol. 35, no. 9, pp. 12796-12809, Sept. 2024, doi: 10.1109/TNNLS.2023.3264740.

 [8] Ruan, Y., & Joe-Wong, C. (2022). FedSoft: Soft Clustered Federated Learning with Proximal Local Updating. *Proceedings of the AAAI Conference on Artificial Intelligence*, *36*(7), 8124-8131. https://doi.org/10.1609/aaai.v36i7.20785

 [9] Dun Zeng, Xiangjing Hu, Shiyu Liu, Yue Yu, Qifan Wang, & Zenglin Xu. (2023). Stochastic Clustered Federated Learning.

---

### Meta-Review · Area_Chair_fgmF · 2024-12-21

**Metareview:**

This paper presents FLAG, an FL method that integrates data and gradient similarities for client clustering, aiming to address challenges posed by heterogeneous client populations. While the reviewers appreciated the clear motivation behind FLAG and its intuitive methodology, they identified several areas for improvement that prevent it from meeting ICLR's high standards. Concerns were raised about the method's novelty, as it largely combines existing techniques without introducing substantial innovation. Scalability and privacy issues were also noted, particularly regarding the potential risk of exposing client data through the transmission of singular values and gradients. Furthermore, the experimental setup relies on custom partitioning methods specifically designed for FLAG, which limits the generalizability of the results. Critical aspects, such as sensitivity to hyperparameters and stronger theoretical or empirical justification, would benefit from significant revision to strengthen the paper’s contributions.

**Additional Comments On Reviewer Discussion:**

The reviewers expressed concerns across multiple aspects of the paper, including its motivation, novelty, scalability, clustering accuracy, handling of data heterogeneity, and the lack of robust theoretical or empirical justification. In the rebuttal phase, the authors addressed some points with point-by-point responses but did not sufficiently engage with all the feedback provided. I would like to thank the reviewers for offering comprehensive and constructive critiques. Overall, the submission falls short of the high standards expected at ICLR.

---

### Decision · Program_Chairs · 2025-01-22

Reject